# Residents’ Attitudes towards Wooden Facade Renovation and Additional Floor Construction in Finland

**DOI:** 10.3390/ijerph182312316

**Published:** 2021-11-23

**Authors:** Markku Karjalainen, Hüseyin Emre Ilgın, Lauri Metsäranta, Markku Norvasuo

**Affiliations:** School of Architecture, Faculty of Built Environment, Tampere University, P.O. Box 600, FI-33014 Tampere, Finland; markku.karjalainen@tuni.fi (M.K.); late.metsaranta@gmail.com (L.M.); markku.norvasuo@tuni.fi (M.N.)

**Keywords:** residents, attitudes, wood, facade renovation, additional floor construction, Finland

## Abstract

To date, studies that provide a comprehensive understanding of residents’ attitudes towards wooden facade renovation and additional floor construction are lacking in the literature. This paper examined these important practices from the perspective of Finnish residents via a questionnaire survey. The 243 responses received highlighted the following: (1) residents’ attitude towards wooden facade renovation and additional floor construction was generally positive; (2) younger and more educated people welcomed these practices more; (3) respondents mostly thought that wooden facade renovation and additional floor construction will increase the attractiveness of residential areas; (4) vast majority were positive about facade renovation, especially with wood; (5) apartment owners welcomed the housing association’s decision to build additional floors to fund the facade renovation; (6) participants assessed the combination of additional floors with outbuildings, followed by additional floor construction alone as the most suitable ways to expand residential areas; and (7) respondents’ attitudes towards all renovation proposals aimed at improving the initial condition of suburban apartments were positive and differed only slightly from each other in terms of popularity. It is believed that this study will provide insights to interested parties, e.g., architects, developers, contractors to better meet users’ needs in the renovation of suburban apartments.

## 1. Introduction

In parallel with the 2050 Energy Roadmap [1], which emphasizes the importance of decarbonizing the growing share of renewable energy and utilizing energy more effectively, Finnish building codes have been adapted and improved accordingly, paving the way for new construction methods to become more energy-efficient [2,3]. The stock of Finnish buildings, most of which were built before the 1990s, was poor in energy efficiency compared to newly constructed buildings, and many were due for renovations [4,5]. About a third of the residential buildings, which make up a significant portion of the Finnish building stock, were apartments built in the suburbs in the 1960s and 1970s and were also in need of renovation [6,7,8].

Suburban apartment buildings of the 1960–1970s were poorly insulated and no energy efficiency targets were set for Finland, which could not be achieved without energy upgrades in those apartments of the period [9]. Energy upgrades to the old building stock should be taken as a way to improve energy efficiency as building stock has been replaced very slowly through new construction, only 1–1.5% per year [10]. The main problems identified in Finnish suburban apartments are poor technical condition, low energy efficiency, lack of equipment, lack of balconies for small apartments, lack of elevators, and unpleasant appearance [11,12]. Additionally, suburbs often have a worse reputation than other residential areas in Finland [12].

Renovating an apartment by improving accessibility and energy efficiency, necessitates a lot of capital and government subsidies [13,14,15,16]. One of the effective methods of financing property development and renovation measures such as facade renovation by increasing energy efficiency and functionality is the construction of additional floors [17]. The terms additional floor, roof, or elevation construction are used when the roof shape of the building changes, the height of the building, and the number of floors increase. Additional floor construction has many advantages such as increasing property owners’ income, having a lower carbon footprint than demolition and new construction, significantly increasing the total floor area without significantly affecting the building’s total energy consumption, and enlivening the image and appearance of the building [18,19].

On the other hand, the materials used in additional floor construction and renovation, e.g., facade renovation should be renewable, recyclable, long-lasting, and their production should consume minimum energy and produce minimum emissions [17,20]. Studies have shown that wood has many advantages over conventional building materials such as steel and concrete, especially in terms of environmental properties, contributing to making it an ideal material for renovation [21,22,23,24,25]. In this sense, timber buildings are considered lower carbon (less fossil fuel-intensive) structures than non-timbered buildings [26,27,28,29], and wood construction represents a lower embodied energy consumption compared to concrete and steel production [30], while concrete production accounts for around 8% of world CO_2_ emissions [31]. Additionally, buildings using steel and concrete embody and consume 12% and 20% more energy, respectively, than timber buildings [32]. Therefore, embodied carbon can contribute up to 80% of the total lifecycle emissions of a residential building [33,34], and the choice of structural materials has a significant impact on embodied carbon [25,35]. On the other hand, using bio-based materials with high carbon storage capacity, e.g., timber, can create a man-made global carbon sink while also reducing CO_2_ emissions associated with construction industry activities [24]. Furthermore, both in manufacturing and on-site construction, steel, and concrete buildings use 7% and 50% more resources than timber, producing 6% and 16% more solid waste [32]. Today, wood structures are becoming popular for tall buildings (over 8-story) due to the rapid growth of engineered wood products such as cross-laminated timber (CLT) [36]. In particular, the construction phase of engineered wood construction can yield significant savings with faster assembly times that are over 50% faster than other alternative materials [37].

On the other hand, the use of timber products strongly supports the European Union’s updated Bioeconomy Strategy [38]; here these products are accepted as a measure to reduce carbon emissions in the construction industry, thus facilitating the transition to a sustainable bioeconomy. Moreover, thanks to their load-bearing capacity and flat roof arrangements, Finnish suburban apartment blocks built in the 1960s and 1970s often allow the construction of additional floors using lightweight structures such as timber construction, which also complies with Finland’s new national fire regulations [17].

Residents’ attitudes towards new construction methods, such as timber facade renovation and additional floor construction, play a critical role in the diffusion of these practices and contribute to the transition to a forest-based bioeconomy in Finland [39]. While there are many studies on various building solutions based on the use of engineered wood products in construction with their technological aspects [40,41,42,43,44,45,46,47,48,49,50], few studies are concentrating on wood as a structural material in buildings from the standpoint of key professionals such as architects, structural engineers, contractors (e.g., [51,52,53,54,55,56]) or users (e.g., [57,58,59,60,61]).

Moreover, there are very limited studies on wooden additional floor construction, and these are not perceptual studies. Among them, Karjalainen et al. [62] scrutinized different phases and advantages of wooden additional floor construction from the perspective of Finnish housing and real estate companies through interviews with professionals involved in these projects, while Soikkeli [17] presented the research project aiming to develop an industrial scale, economical and efficient concept for renovating, expanding, and adding floors to apartment blocks. Therefore, there is no study in the literature on the perception of the residents about wooden facade renovation and additional floor projects.

In this study, the following main research questions were identified:How do residents react to the timber facade renovation and additional floor construction?How do wooden facade renovation and additional floor construction affect the attractiveness of residential areas?What are the participants’ preferences regarding additional construction issues?How do residents perceive different wood renovation alternatives?

Overall, this research aimed to provide a representative understanding of Finnish residents’ attitudes towards wooden facade renovation and additional floor construction through a questionnaire survey. In doing so, this study sought to identify the main parameters behind the effect of wooden facade renovation and additional floor construction on the attractiveness of residential areas, residents’ preferences for additional construction, attitudes of apartment building owners towards their own housing company’s wooden additional floor project to finance facade renovation, residents’ attitudes towards different wooden renovation alternatives, as well as residents’ attitude towards wooden facade renovation and wooden additional floor construction in Finland. The results are believed to provide critical stakeholders with a roadmap for renovating suburban apartments to better meet residents’ needs. This will contribute to increasing the acceptability and attractiveness of wooden facade renovation and additional floor construction for Finland and other countries. It is worth noting here that resident willingness is an important factor in speeding up or slowing down the successful implementation of comprehensive renovation [63].

In this research, wood or timber refers to engineered timber products [64,65] such as cross-laminated timber ((CLT) a prefabricated multi-layer EWP, manufactured from at least three layers of boards by gluing their surfaces together with an adhesive under pressure), laminated veneer lumber ((LVL) made by bonding together thin vertical softwood veneers with their grain parallel to the longitudinal axis of the section, under heat and pressure)), and glue-laminated timber (glulam) ((GL) made by gluing together several graded timber laminations with their grain parallel to the longitudinal axis of the section)).

The remainder of this paper was structured as follows: First, an explanation of the research methods used in the study was provided. This was followed by findings of residents’ attitudes towards wooden facade renovation and additional floor construction in Finland. In Section 4, the discussion part was given. Finally, the results of the study were presented with suggestions for future research.

## 2. Research Methods

This study was carried out mainly as a literature review including international peer-reviewed journals and similar research projects, supported by materials collected during “the Wood at Visibility at the Tampere University”—a project that is part of the Ministry of the Environment’s Growth and Development from Wood support program involves cross-sectional data from the Pukinmäki–Savela area in the City of Helsinki (Finland). The main emphasis of the project was to increase the competitiveness of wooden apartment buildings and to demonstrate the potential of wood in energy enhancement and the construction of additional floors in suburban apartments.

In the survey, a combination of postal and online questionnaire surveys was administrated, in which online inquiry and paper forms contained the same questions in principle. The main population of the material was 800 Finnish-speaking people aged 18–70 in the postcode 00720, or Pukinmäki–Savela area (Figure 1), which has the permanent address of the region according to the population information system. The sample was chosen randomly, the responses were handled anonymously, and no personally identifiable data was collected or used in the analysis stage. The Pukinmäki–Savela area was chosen because it represents a typical Finnish suburban district in terms of building stock and density and is suitable for suburban development research on this basis.

The mailing of the surveys started on 18 May 2020. The first responses to the online survey arrived on 25 May; based on which it can be assumed that the paper forms reached residents by mail the same day. Forms for initial mail submission, return deadline was 1 June, with residents just one week left to respond. Response time was shorter than planned due to mail delay. Initially, a total of 149 responses were received to the survey. Responses were distributed with 110 responses on paper forms and 39 responses in an online questionnaire. Reminder round questionnaires were delivered to the post office on June 10. The actual deadline for response was abandoned and instead, respondents were asked to return a completed questionnaire one week after it arrived. This was to avoid the impact of the mail delay response time. Paper surveys for the reminder round reached residents based on online survey responses on 15 June. During the reminder round, 94 new people arrived, 67 on paper and 27 online. A total of 243 responses were received for the entire questionnaire, which corresponds to a response rate of 30%.

At the beginning of the survey, the topic was carefully explained to allow residents to form ideas on how to create and answer questions about it. Participants were first briefly introduced to the topic with an explanatory text and a series of pictures on the economic benefits of additional floor construction, followed by actual questions about wooden facade renovation and additional floor construction.

The questionnaire focusing on facade renovation and additional floor construction was divided into 6 parts as seen in Table 1.

**Table 1 ijerph-18-12316-t001:** The questionnaire used in this study.

#	Part	Explanation
1st	A	After the participants were introduced to the topic with a series of pictures from the questionnaire, they were asked to describe their attitude towards wooden facade renovation at a general level. A Likert-type scale was used ((1) negatively to (5) positively). Here “I don’t know” option was also provided.
2nd	B	Using the same Likert-type scale in Part A, they were asked to explain their attitude towards additional floor construction.
3rd	C	Residents were asked to describe their opinions on the combination of wooden facade renovation and additional floor construction. Another Likert-type scale was used ((1) completely disagree to (5) completely agree). Here “I don’t know” was provided, too.
4th	D	Participants were asked about their preferences for additional construction (e.g., with an additional floor, new outbuildings) in residential areas.
5th	E	Respondents who own an apartment were asked to indicate how they would react if their housing association decided to build an additional wooden floor to finance the renovation of the facade by using the same Likert type scale in Part A and B to improve the features that respondents considered to be a failure in a suburban apartment building. Participants were allowed to provide feedback on facade renovation and additional floor construction in open responses.
6th	F	Residents surveyed were presented with a series of six pictures of wood facade renovation and additional floor construction projects implemented in a typical Finnish suburban apartment building in the 1970s, as seen in Figure 2. The initial situation of the suburban apartment building before the renovation was shown first in the series picture (Image 1). Images 2–5 of the series were visualization images of possible wooden facade renovation and additional floor implementations, all of which are based on the typical suburban apartment building of Image 1. Participants were asked to look at pictures of facade renovation and additional floor construction and then rate them. School grading scale (from 4 (I don’t like the look of the building and I have a negative attitude like this building) to 10 (appearance of the building is beautiful and I welcome this building)) was employed.

Since most of Finland’s suburbs were built in the 1970s [66], the base image (Image 1/initial situation) was chosen to represent the typical apartment building of that era and consider the suitability of additional floor construction. Image 1, a 4-story suburban apartment building, was meant to be ordinary and was rendered anonymous by an image editing program. The concrete-framed building had a flat roof, and its first floor was the so-called “above ground basement”, i.e., a floor where no dwellings are located, only housing support functions, such as bicycle storage (and sometimes garages). The facades had a painted concrete surface up to the balconies. To enliven the facade, part of the concrete facade was in a different color.

The series of images aimed to find the participants’ attitudes towards wooden facade renovation with additional floors and compare them to an unrenovated suburban apartment. These visuals were also deemed necessary for the participants to understand what the construction methods meant. While facade renovation and additional floor construction alternatives were produced, it was tried to present as many options as possible so that the participants did not have a single idea about the construction method. If only one design solution had been made for the construction site, the participants could have considered wooden facade repair and additional floor construction could only produce solutions such as this plan. On the other hand, the plans were kept in moderation so as not to arouse too strong feelings towards the architecture.

The goal in Image 2 (style with minimal solution) renovation alternative was to keep the building height suitable for low-rise cityscape, so there was only one wooden additional floor. This design solution was the most subtle of the options, but at the same time brings the least additional housing to the building. The wooden facade was renovated, and the building was upgraded with an additional floor. The spaces between the balconies were left lighter than the other wooden parts. The bright part continued up to the top floor balconies that gave life to the facade of the building. The first layer remained on a concrete surface.

In the design of Image 3 (modern look), the aim was to get the participants’ views on a powerful reimagining of a renovation solution, in which the old building was modernized beyond recognition. The building was raised one and a half floors, and additional wooden floors, facades, and balconies were converted to wood. The entire building character was transformed into a playful asymmetrical layout with a more modern aspect and a more attractive new entrance, enlivening the facade of the building with the asymmetrical arrangement of the balconies.

The objective of the remediation alternative in Image 4 (inspiration from suburban architecture) was to get the views of the respondents in the traditional suburban landscape fitting in shape and color the plan. The architecture of this proposal largely followed the baseline of the formal language of a suburban apartment building with a simple elevator solution. The wooden facade was renovated, the upper part of the building was retracted, and the second upper floor was raised with two additional floors in harmony with the appearance of the lower ones. The retracted additional layer was separated by the other paste color. The application was facilitated by adding new elevators that were completely outside the old building mass. The dark color at the entrance continued along with the protruding elevators to the top floor. The other wooden parts of the facade were painted white. The coloring contrast separated additional elements of a different nature from the building.

In Image 5 (implementation of the gable roof), the aim was to get respondents’ views on the additional floored structure of the gable roof and to be embedded in the building’s collective elevator solution. The building was raised with two wooden additional floors. The upper additional floor of the building was placed on a partially steep gable roof. New elevators were installed in the building, partially coming out of the facade. The facade cladding was a combination of vertical and horizontal cladding, while the facade surface of the top additional floor was machine-sealed sheet metal. Building coloration ranged from gray to blue tones.

Finally, in Picture 6 (wood in main parts), the building was elevated with two additional wooden floors and covered with wooden horizontal cladding. The building had a slightly flat roof, although, at an angle, the result was unobtrusive. The new elevators were located partly outside the building and partly inside the old mass. The construction of the two upper floors was narrower horizontally than the lower floors and allowed the additional floors to stand out from the old ones. The facade cladding was left to color the natural wood. The new apartments on the additional floors mainly had French balconies. Additional floor window openings and balcony solutions were deliberately made to highlight the architectural appearance of the old building mass.

## 3. Findings

### 3.1. Residents’ Attitude towards Wooden Facade Renovation

Surveyed residents’ attitude towards wooden facade renovation was generally positive (the total occurrence of “positively” and “partially positively” options) (91%), with a negligible minority of respondents having negative perception (1%)—see Figure 3. According to the responses, it was observed that while the positive attitude was generally preserved in the age groups, “positive” attitude was replaced by “partially positive” in the older age groups. This may be because young people have a generally positive appreciation of wood for its health and sustainability impacts in the housing context [67,68]. Moreover, the fact that higher age can deter people from investing in activities due to shorter lifespans, health problems, diminishing abilities, and financial constraints may have contributed to this attitude [69]. They thus tend to leave energy-efficient retrofits such as facade renovation to younger generations. Here, the “I do not know” option had low occurrences compared to others. 

With the increase in the level of education, the attitude towards facade renovation evolved towards a more positive one. This may be because higher educated people have relatively higher awareness and prior knowledge of the advantages of wooden facade renovation [70]. The gender of the respondents did not affect attitudes. The owner of detached houses was slightly more positive about facade renovation than residents of apartments. Those who live in a small house were probably more connected to existing housing areas and therefore view suburban development more positively.

### 3.2. Residents’ Attitude towards Wooden Additional Floor Construction

As shown in Figure 4, respondents were also in favor of wooden additional floor construction (the total occurrence of “positively” and “partially positively” options) (85%), while a little more resistance (4%) was observed against the construction of additional floor compared to the facade renovation. In the open answers, the attitude towards additional floor construction was mainly reflected as positive comments. When the effect of background variables on the attitudes of building additional floors was examined in more detail, it was seen that it had a parallel effect with the age and education level of the participants, as in the case of wooden facade renovation. As the education level increased, the positive attitude increased. This can be attributed to the fact that higher education is assumed to be linked to greater abilities to gather information, understand it, and put it into action [69], and highly educated people e.g., university graduates may further explore the possibilities of new technological systems [71]. In addition, as the age group increased, the total percentage of positive and partially positive attitudes decreased, excluding the group under the age of 30. On the other hand, males had a more positive perception than females.

### 3.3. Effect of Wooden Facade Renovation and Additional Floor Construction on the Attractiveness of Residential Areas

As seen in Figure 5, while 67% of the participants thought that wooden facade renovation and additional floor construction would increase the attractiveness of residential areas, 17% thought that this would not affect the situation. Although females (68%) and males (64%) thought that these changes would increase the attractiveness of the residential areas, the ratio of females who did not express their opinion (18%), that is, chose the option “I do not know”, was higher than males (13%). As the age group increased, the view that the attractiveness of the areas would increase turned into a negative one and the rate of neutral perception increased (i.e., the dark blue ratio dropped from 85% (for those under 30) to 50% (for those over 60)), but this trend reverses as the education level increased ((i.e., the dark blue ratio increased from 52% (for primary school graduates) to 71% (for university graduates)) as highlighted in Figure 5. Similar to residents’ attitudes towards wooden facade renovation as shown in Figure 3, the negative impact of age (e.g., [69]) and the positive impact of education (e.g., [70]) were reflected in the residents’ views on the attractiveness of the residential areas due to wooden facade renovation and additional floor construction.

### 3.4. Residents’ Preferences for Additional Construction

Residents’ attitude towards wooden facade renovation and additional floor construction for their residential areas was generally positive (the total occurrence of “completely agree” and “partially agree” options) (89%), with a minority of respondents having negative perception (5%)—see Figure 6. Less than two-thirds (65%) of respondents thought that the choice of materials was at least partially unimportant and that the beauty of architecture was at least partially more important than the choice of materials. Just over a fifth (21%) of participants rated the importance of wood as being of primary importance when performing facade renovation and additional floor construction. When the building reached the age of renovation, survey respondents felt that wood can be inserted into facade renovation and additional floors. The open answers indicated that old buildings should be built on additional floors rather than demolished. Residents of the Pukinmäki–Savela area were therefore surprisingly positive about bringing a better wood renovation to their area of residence.

As seen in Figure 7, additional floor construction had a lot of potential because more than half (55%) residents opted for this method. The most popular option for additional construction was the additional floor construction with the new outbuilding, supported by one-third of the respondents, followed by additional floor construction with 22% occurrence. Survey respondents perceived the demolition of old buildings and the construction of new, more efficient buildings as a more viable solution than simply building new outbuildings. Males were somewhat more positive about the demolition reorganization. Feedback from open answers showed that residents feared that their residential areas would be built too much; this can partly explain the popularity of demolition.

### 3.5. Attitudes of Apartment Building Owners towards Their Own Housing Company’s Wooden Additional Floor Project to Finance Facade Renovation

Surveyed apartment building owners’ attitudes towards wooden additional floor were generally positive (the total occurrence of “positively” and “partially positively” options) (63%), while about one-fifth (21%) of respondents had a negative view of this—see Figure 8. This high rate of positive attitudes may be due to the economic advantages of additional floor construction, for example its revenues can be used to finance the renovation of the existing property (renovation of an elevator to improve the accessibility and commercial conditions of the building) [17,62].

### 3.6. Residents’ Attitudes towards Different Wooden Renovation Alternatives

Participants’ attitudes towards all types of wooden facade improvements were positive as seen in Figure 9. While the original suburban apartment (Image 1) received a score of 5.44 on a scale of (4–10), the average of the proposed projects was 7.65. Facade renovation and additional floor construction proposals were consistently strong and did not differ significantly in popularity. The mutual order of their popularity among residents, from most to least: Image 5 (implementation of the gable roof), Image 2 (style with minimal solution), Image 3 (modern look), Image 4 (inspiration from suburban architecture), and Image 6 (wood in main parts).

Figure 10 shows the distribution of the ratings in the series by Image. The concrete suburban apartment building (Image 1) was rather weaker than the proposals’ scores. The vast majority (>80%) had at least a partially negative opinion of the apartment at baseline (at least partially negative = less than 7, neutral attitude = 7, at least partially positive = more than 7). The gable roof design proposal (Image 5) received the largest share, which can also be seen in the rating distribution. As a generalization, it can be said that about half of the respondents welcome at least partially solution options (Image 2–6); one-quarter neutral and about one-sixth are at least partially negative. Respondents justified their ratings with open answers, in which coloring, bare concrete surface, above-ground basement, architecture (boredom/box-like), flat roof, lack of balcony glazing, and lack of elevators were considered as the most major problems.

## 4. Discussion

Due to the lack of literature, it has not been possible to conduct a comprehensive discussion to provide information on the similarities and differences of Finnish residents’ attitudes with examples from other countries, as reported by Karjalainen et al. [62] in the study on wooden additional floor construction from the standpoint of Finnish housing and real estate companies.

According to the survey results, residents’ attitudes towards wooden facade renovation and additional floor construction were generally positive. This finding resembled the finding in a Chinese study on the issues and challenges of implementing comprehensive renovation at aged communities [72], where residents were willing to accept basic renovation measures. 

Our results also showed that younger and more educated people had more positive attitudes towards these applications. Similarly, a Slovenian study based on analysis of more than 1000 single-family-house owners by Hrovatin and Zorić [69] reported that higher homeowners’ age may be the main barrier to facade refurbishment, and the high level of education positively affected the attitudes of homeowners towards facade renovation. Likewise, the positive effect of education [73,74,75] and the negative impact of age [70,73,74,76] were demonstrated in many studies on renovation or energy-saving measures. 

Most of the surveyed residents considered that wooden facade renovation and additional floor construction would increase the attractiveness of residential areas. This can be associated with the recommendation by Viholainen et al. [77] that the use of wood on surfaces (also in non-timber urban buildings) should be increased to generate attractive soundscapes. Similarly, Sandberg et al. [78] reported that the use of prefabricated wooden elements for the sustainable renovation of residential building facades increases the attractiveness of the environment and housing blocks according to the surveyed residents. Additionally, increased living comfort and thermal comfort were underlined in the study by Hrovatin and Zorić [69] on the renovation of the building envelope.

Respondents in the survey generally approached the facade renovation in residential areas positively and thought that it should be done with wood. The emphasis on wood here can be associated with the emphasis of the residents on well-being, aesthetic, and environmentally friendly features of wood as a construction material and timber frame houses, as stated in the study by Gold and Rubik [59].

The vast majority found the renovations with wooden facades and additional floors suitable for the apartments in the renovation age. It should be noted here that the age and service life of the building elements were important reasons for the decision to renovate in many studies [79,80].

Apartment owners welcomed the housing association’s decision to implement additional floor construction to finance the facade renovation. On the other hand, participants assessed the combination of the construction of additional floors with outbuildings, followed by the construction of an additional floor alone as the most popular way to expand their residential areas. Moreover, open answers indicated that because it is desirable to preserve environmental spaciousness in residential environments, residents prefer to have a stock of buildings that will expand the existing rather than concentrate.

Thus, additional construction should be moderate, as the survey respondents’ biggest fear was that residential areas would be built too densely. Additionally, respondents did not prefer additional construction taking up space in park-like and unspoiled areas, which was why additional floor construction is in many situations the best option. Finally, the attitudes of the participants towards all kinds of wooden facade improvements (e.g., style with minimal solution, with gable roof) were positive and differed only slightly from each other in terms of popularity. 

According to the survey results, considerations such as architectural boredom, bare concrete surface, flat roof, lack of elevator were considered as the most important problems of suburban apartments among the participants. Proposed solutions to these problems are shown in Figure 11.

## 5. Conclusions

This study aimed to present a representative analysis of Finnish suburban residents’ attitudes towards wooden facade renovation and additional floor construction, which were proposed as solutions for neighborhood development. In doing so, this study sought to identify the main parameters behind the effect of wooden facade renovation and additional floor construction on the attractiveness of residential areas, residents’ preferences for additional construction, attitudes of apartment building owners towards their own housing company’s wooden additional floor project to finance facade renovation, residents’ attitudes towards different wooden renovation alternatives, as well as residents’ attitude towards wooden facade renovation and wooden additional floor construction in Finland.

This study can be used in the renovation of suburban apartments as a design guide for interested parties (e.g., architects, contractors, developers, policymakers) emphasizing the great potential of wooden facade renovation and additional floor construction. In this sense, a better understanding of suburban housing expectations and the possibilities of future generations can be achieved in Finland.

Moreover, wood construction, and the forest-based industry in general, is not only an important part of the Finnish economy but also the key to the transition to a bio-based economy. However, a new generation that is considered more environmentally friendly and follows a radically different lifestyle from previous generations is on the rise. From the perspective of the forest-based industry, this presents a great opportunity to place wooden practices in the minds of future generations of stakeholders, not only as a way to build in the future, but also to reposition itself as part of the solution to overcome the challenges of environmental and sustainability issues. It is thought that this research will contribute to this important issue.

In conclusion, this paper aimed to fill the gap in the literature regarding Finnish suburban residents’ attitudes towards wood facade renovation and additional floor construction and to generate insights into the residential renovation market so that communication measures can be adapted to users more effectively. In this sense, supporting positive market development, making people aware of the needs and benefits of energy savings from building renovations, creating green retrofit organizations and professionals, providing various types of financial support for building renovation, further investing in research and development activities for this, and promoting new modes of energy contracts are critical.

Future research should explore more models of analysis for the qualitative factors (e.g., human-centric, personal, and contextual parameters) influencing residents’ decisions about building renovation, with emphasis on cases where the rise of new policies has focused on the social dimension of renovation rather than technical improvements for buildings. Future work may also include other background variables (e.g., residents’ income level, past residence experience) to get more insight into residents’ attitudes towards wooden facade renovation and additional floor construction. Policy measures need to be introduced to strengthen the role of residents in the building renovation process. In addition, similar studies can be carried out in other Scandinavian countries and the subject can be enriched with comparative analyzes.

## Figures and Tables

**Figure 1 ijerph-18-12316-f001:**
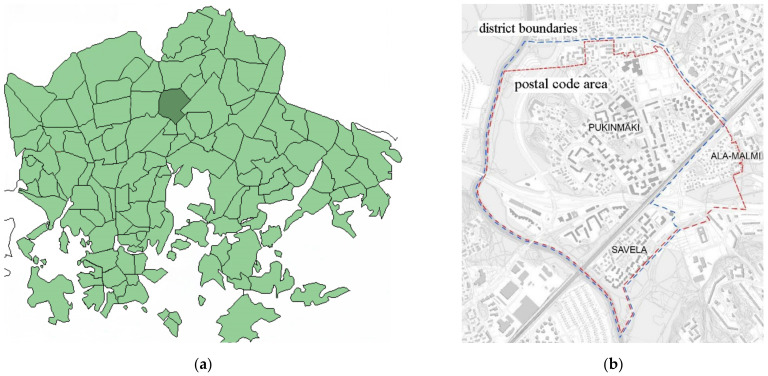
Pukinmäki–Savela area in the City of Helsinki (Finland); (**a**) position of Pukinmäki–Savela area within Helsinki subdivision map; (**b**) district boundaries and postal code area of Pukinmäki–Savela area.

**Figure 2 ijerph-18-12316-f002:**
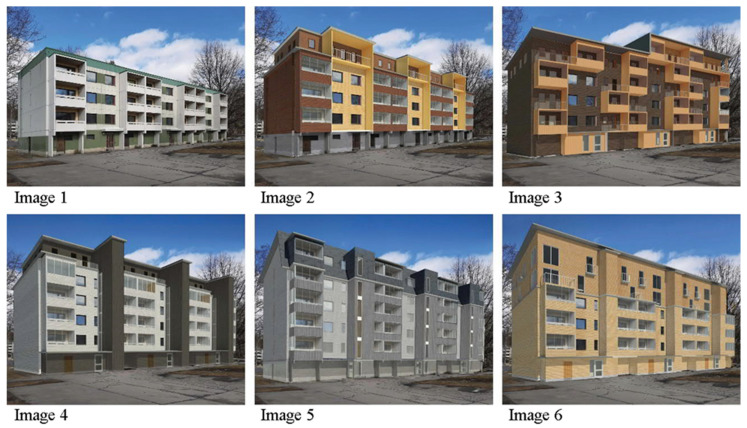
Original concrete apartment building (**Image 1**) and proposed renovation alternatives (**Image 2**–**6**).

**Figure 3 ijerph-18-12316-f003:**
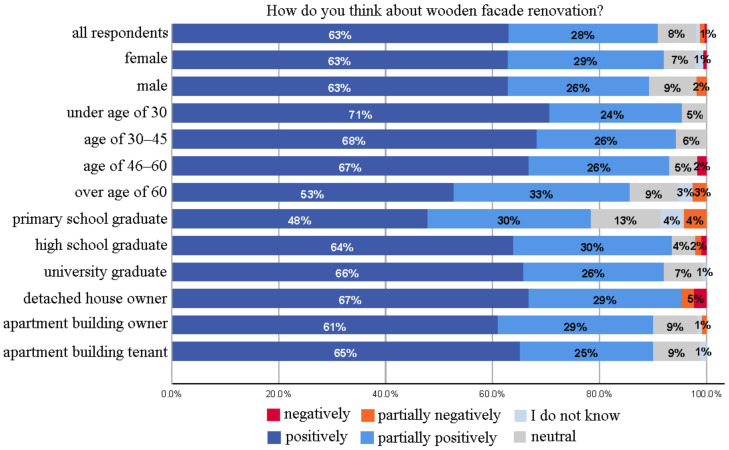
Attitudes towards wooden facade renovation.

**Figure 4 ijerph-18-12316-f004:**
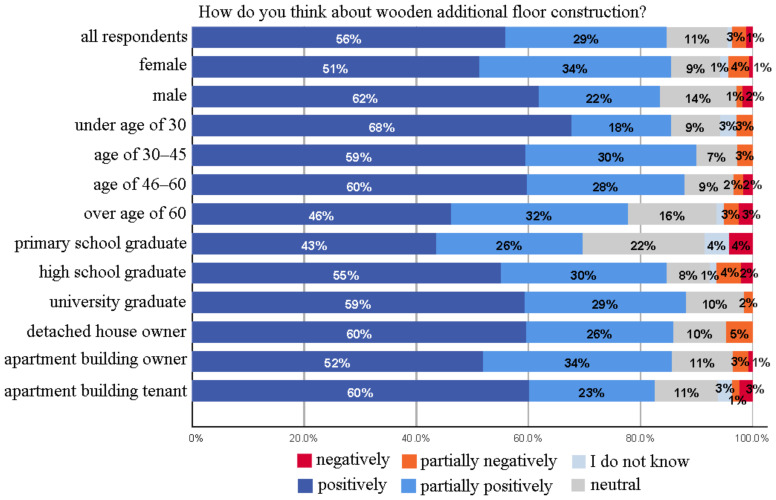
Attitudes towards wooden additional floor construction.

**Figure 5 ijerph-18-12316-f005:**
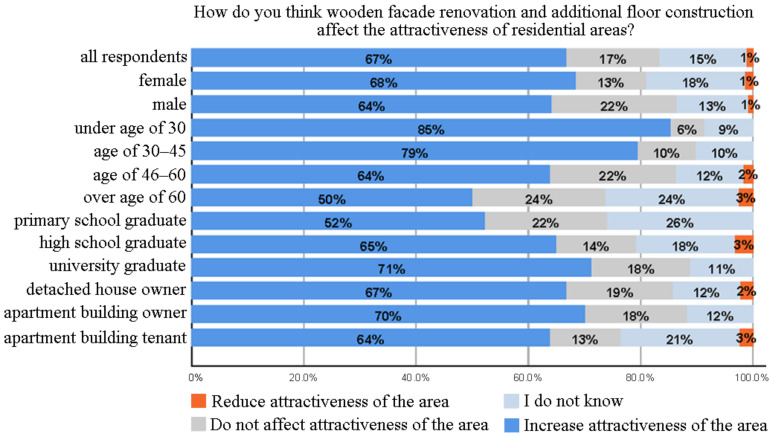
Effect of wooden facade renovation and additional floor construction on the attractiveness of residential areas.

**Figure 6 ijerph-18-12316-f006:**
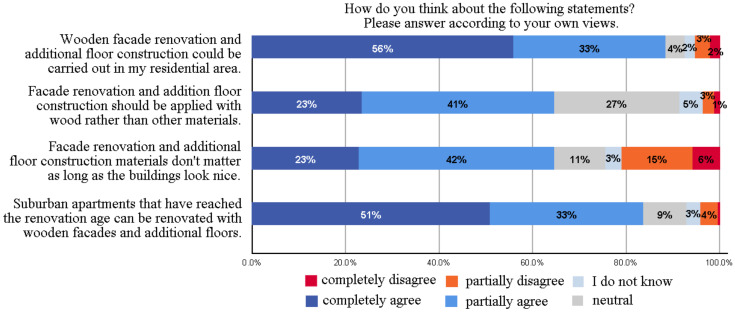
Preferences for additional construction.

**Figure 7 ijerph-18-12316-f007:**
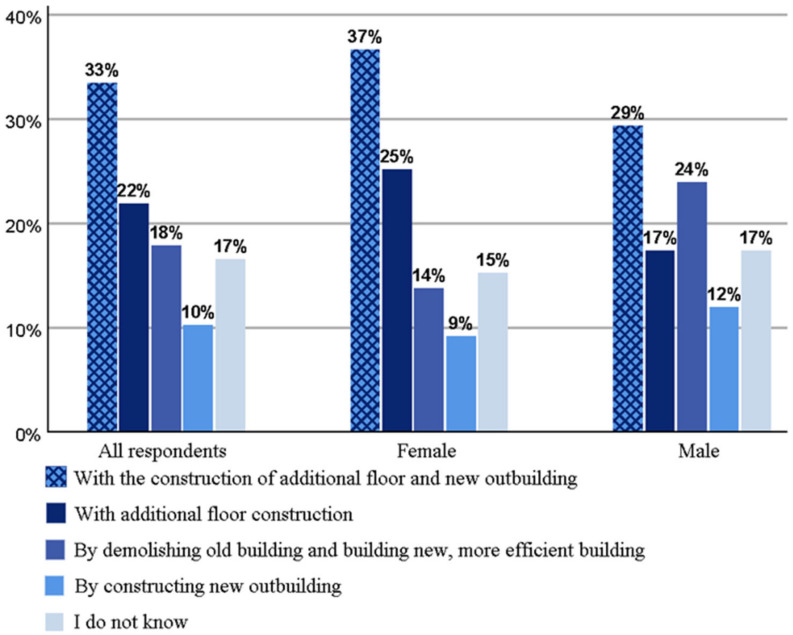
Preferred ways to implement additional construction in residential areas.

**Figure 8 ijerph-18-12316-f008:**
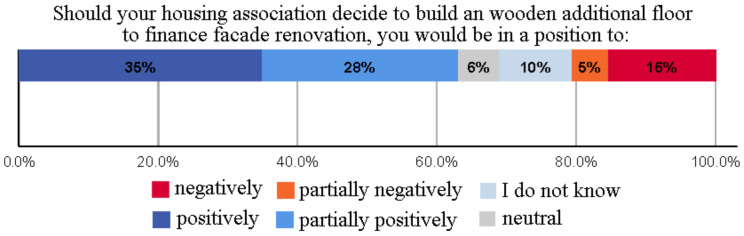
Attitudes of apartment building owners towards their own housing company’s wooden additional floor project to finance facade renovation.

**Figure 9 ijerph-18-12316-f009:**
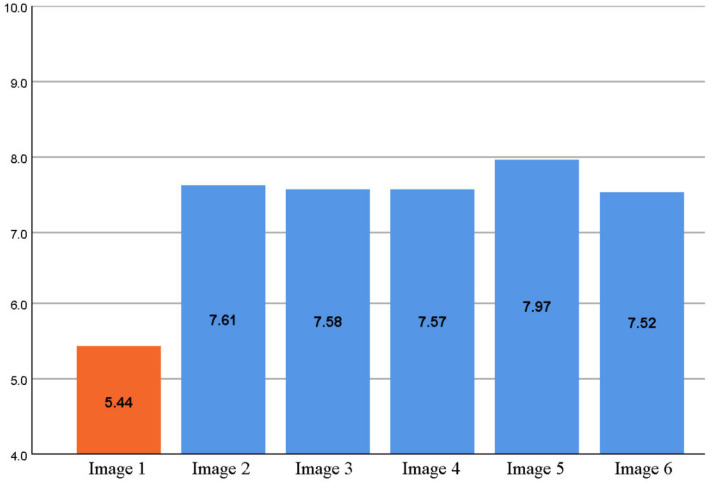
Residents’ attitudes towards different wooden renovation alternatives.

**Figure 10 ijerph-18-12316-f010:**
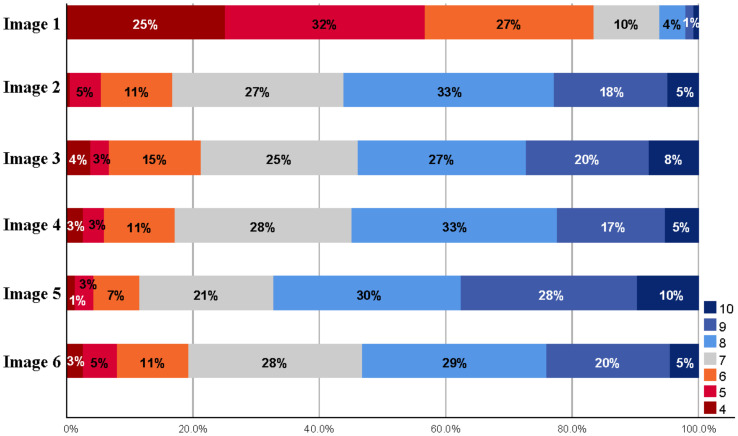
Distribution of grades in the proposed renovation alternatives.

**Figure 11 ijerph-18-12316-f011:**
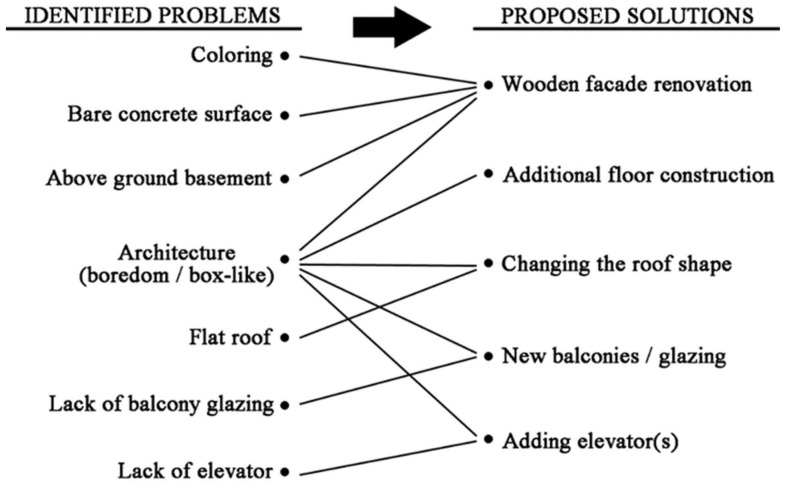
Identified problems of Finnish suburban apartments and their proposed solutions.

## Data Availability

Not applicable.

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
