# Peer review of "Residents’ Attitudes towards Wooden Facade Renovation and Additional Floor Construction in Finland"

_ijerph, 2021, doi:10.3390/ijerph182312316_

Round 1

Reviewer 1 Report

This paper aimed to present a representative analysis of Finnish suburban residents' attitudes towards wooden facade renovation and additional floor construction. It provides insights to interested parties, e.g., architects, developers, contractors, to better meet users' needs in the renovation of suburban apartments.

The structure of the manuscript is well. The manuscript has some of its technical merits. The topic is very interesting. It may fall into the Journal about Construction. Anyway, several questions may need to pay attention to:

  1. There are some typos in the paper. It is suggested that the manuscript be submitted after carefully checking before publication.
  2. In the paper, the authors listed the main research questions. However, the main contributions of the paper have not been mentioned. Recommend the authors should highlight it.
  3. In section 1, the last paragraph, the authors haven't described the whole structure of the paper. Recommend the authors should add it.
  4. There are two images in Figure 1. The author ignored the sub caption of each image. Recommend the authors need to add the sub caption of each image.
  5. The authors describe the questionnaire from lines 157 to 183. If the author provides a table, it will be better.
  6. In section 3, the authors only provide the findings from Figures. However, there are very few explanations for those findings. Recommend the authors should give an essential explanation.
  7. In section 3.2, line 265, the authors mentioned," as the age group increased, the positive attitude de-265 creased." However, the positively rate is an increment from the age of 30-45 to the age of 46-60. Please check it.
  8. In section 3.3, line 275, the authors mentioned," females took a more neutral stance." How to get the statement? Please provide the evidence.
  9. There is only one index (rate) in the finding. The authors might explore much more indexes. It will be better.
  10. In section 4, the discussion can be improved. The authors should point out the disadvantage of the proposed algorithm or the comparison of enough previous literature. The authors need to extend the discussion.
  11. In section 5, the conclusion is very short and simple. The author might enrich your conclusions. It might also suggest future research.

Hopefully, this will help in the revision of the manuscript.

Reviewer 2 Report

In my opinion the paper is very clear and correctly writed.

Objectives, metodoloogy and results are clearly presented.

Conclusions are coincise but very clear

Reviewer 3 Report

The article examines an important topic that has not been studied. This research explored Finnish suburban residents’ attitudes towards wooden façade renovation and additional floor construction through a questionnaire survey.

The study can benefit from addressing the issues identified below, which I consider critical weaknesses of the study.

  1. In line 57, researchers stated that wood has many advantages over conventional buildings materials. In relation to this, please provide more detailed and specific aspects of the advantages of using wood compared to conventional buildings.
  2. In lines 157-183, please provide a clear list of questions through numbering, tables, etc. It is unreadable in the current manuscript.
  3. In line 194, please clearly provide the criteria of the proposed renovation shown in images 2-5 before explaining each of them.

Round 2

Reviewer 1 Report

The authors have improved the manuscript according to most of the points.